# *MYCN* Amplification, along with Wild-Type RB1 Expression, Enhances CDK4/6 Inhibitors’ Efficacy in Neuroblastoma Cells

**DOI:** 10.3390/ijms24065408

**Published:** 2023-03-12

**Authors:** Piergiuseppe De Rosa, Federica Severi, Suleman Khan Zadran, Marco Russo, Sara Aloisi, Alberto Rigamonti, Giovanni Capranico, Giorgio Milazzo, Giovanni Perini

**Affiliations:** Department of Pharmacy and Biotechnology, University of Bologna, Via Selmi 3, 40126 Bologna, Italy

**Keywords:** Neuroblastoma, MYCN, RB, E2F, palbociclib, ribociclib, CRISPRi, ΔCDK

## Abstract

Neuroblastoma (NB) is one of the primary causes of death for pediatric malignancies. Given the high heterogeneity in NB’s mutation landscape, optimizing individualized therapies is still challenging. In the context of genomic alterations, *MYCN* amplification is the most correlated event with poor outcomes. MYCN is involved in the regulation of several cellular mechanisms, including cell cycle. Thus, studying the influence of MYCN overexpression in the G1/S transition checkpoint of the cell cycle may unveil novel druggable targets for the development of personalized therapeutical approaches. Here, we show that high expression of E2F3 and MYCN correlate with poor prognosis in NB despite the RB1 mRNA levels. Moreover, we demonstrate through luciferase reporter assays that MYCN bypasses RB function by incrementing E2F3-responsive promoter activity. We showed that MYCN overexpression leads to RB inactivation by inducing RB hyperphosphorylation during the G1 phase through cell cycle synchronization experiments. Moreover, we generated two *MYCN*-amplified NB cell lines conditionally knockdown (cKD) for the RB1 gene through a CRISPRi approach. Indeed, RB KD did not affect cell proliferation, whereas cell proliferation was strongly influenced when a non-phosphorylatable RB mutant was expressed. This finding revealed the dispensable role of RB in regulating *MYCN*-amplified NB’s cell cycle. The described genetic interaction between MYCN and RB1 provides the rationale for using cyclin/CDK complexes inhibitors in NBs carrying *MYCN* amplification and relatively high levels of RB1 expression.

## 1. Introduction

Neuroblastoma (NB) is the most common pediatric extracranial solid tumor. As one of the first causes of death for children in the first five years of life, it represents 13% of overall children’s cancers [1]. NB arises from neural crest progenitors, which give origin to sympathetic ganglia and/or adrenal glands. As a consequence of that, the primary tumor is frequently located in the abdomen or the adrenal gland [2]. Although NB is a highly heterogeneous cancer, its most common genetic alteration is the MYCN gene amplification, found in up to 50% of high-risk cases [3]. 

MYCN encodes for the transcription factor N-Myc, known to be a crucial player in driving tumorigenesis by sustaining growth through cell cycle genes positive regulation. Among these, N-Myc drives the expression of the cyclin-dependent kinase 4 (CDK4), the minichromosome maintenance protein (MCM), the Myb-related protein B (MYBL2), the serine/threonine-protein kinase cell cycle checkpoint kinase 1 (CHK1), the inhibitor of DNA binding 2 (ID2), and S-phase kinase-associated protein 2 (SKP2) [4].

The molecular mechanisms leading to aberrant cell proliferation and tumor formation in NB are poorly understood. In physiological conditions, the cell cycle checkpoints are precisely regulated by multiple protein families, including cyclins, cyclin-dependent kinases (CDKs), and CDK inhibitors (CKIs) [5]. On the other hand, these checkpoints become necessarily dysregulated for tumor initiation and maintenance, and driver mutations in proto-oncogenes and tumor suppressors genes are responsible for the malfunctioning of the cell cycle machinery.

In the context of the G1/S transition checkpoint, the retinoblastoma protein (RB) and E2F transcription factors play a fundamental role in cell cycle progression [6]. During the early G1 phase, RB interacts and blocks E2F transcriptional activity. Proceeding through the late G1 phase, RB is phosphorylated by cyclin/CDK complexes on up to 15 serine and threonine amino acid residues [7]. This event drives structural changes in RB, resulting in E2F dissociation [8,9]. The E2F gene family encodes for eight transcription factors (E2F1-8) involved in cell cycle progression. However, only E2F1, E2F2, and E2F3 are formal transcriptional activators, and their principal activity is to drive the transition from G1 to the S phase of the cell cycle. At the end of the G1 phase, E2F1-3 factors promote the expression of S genes and the progression of the cell cycle. Among the E2F activators, E2F3 positively regulates proteins involved in the recognition and utilization of replication origins, such as origin recognition complex subunit 1 (ORC1), cell division cycle 6 (CDC6), and mini chromosome maintenance (MCM) as well as cyclin E and CDK2. The accumulation of CDC6, cyclin E, and CDK2 allows the transition to the S phase [10,11].

Overexpression of E2F1 and E2F3 in NB correlates with poor prognosis, independently from *MYCN* amplification. Patients expressing high levels of E2F1/3 showed increased expression of genes involved in G1/S transition [12]. Parodi et al. recently found that overexpression of E2F3 is a hallmark of impaired survival in stage 4S NBs, typically characterized by spontaneous regression [13]. These findings have been obtained by analyzing large datasets of RNAseq; however, the role of these transcription factors in NB biology is still unclear. Recently, the role of E2Fs in NB was further investigated. Studying the effect of the silencing and the pharmaceutical inhibition of the histone demethylase KDM6B in *MYCN*-amplified NB cells, D’Oto et al. found a relevant reduction in cell viability and proliferation, most likely due to the downregulation of MYCN and E2F transcriptional signatures [14].

Among the most important regulators of the cell cycle, MYC, encoding for c-Myc protein, is known to be activated by the E2Fs and vice-versa [15]. MYC induces cell cycle progression by different mechanisms. On the one hand, it directly activates cyclin/CDK complexes, which hyperphosphorylate RB proteins and let the E2Fs be active [16,17]. On the other hand, MYC antagonizes CDK inhibitors such as CDKN1A (p21) and CKDN1B (p27) [18]. MYCN is one of the MYC paralogue genes and shares several features with MYC, such as heterodimerization with the partner protein MAX, binding to the same DNA regions (E boxes), and interaction with the same set of kinases, transcription factors, and chromatin remodelers [19]. 

Recent studies showed that E2F proteins cooperate with N-Myc to drive the oncogenic phenotype [12,20,21]. In fact, E2F1-3 activity is deregulated in most cancers, mainly by their overexpression or in the context of RB1 mutations [6]. In retinoblastoma, the presence of wild-type RB1 is unexpectedly observed in 2.7% of cases [22]. Notably, half of the RB1 wild-type retinoblastoma cases analyzed in this study were characterized by *MYCN* amplification which was instead absent in most RB1 mutated tumors (97.3%). Histologically, this specific subgroup of retinoblastoma shares more similarities with *MYCN*-amplified neuroblastomas than classical retinoblastomas, but molecular mechanisms driving its onset have only begun to be explored [23,24]. Interestingly, Singh et al. discovered that in explanted developing human retinae, N-Myc enhances the expression of CCND2 and CDK4 as MYC does in fibroblasts, leading to phosphorylation and inactivation of RB [24,25]. 

This last line of evidence supports the hypothesis that a high dosage of MYCN in the cell may overcome RB function to promote an E2F-driven oncogenic transformation in NB. To test this hypothesis, we generated NB cell lines in which the expression of RB1 was modulated using the CRISPRi technique and evaluated the cell behavior using cell biology assays and E2F-responsive luciferase vectors as a function of MYCN. Moreover, we aimed to provide additional information about the cell pathway driving the sensitivity of the NB to CDK4/6 inhibitors, asking if the expression of RB1 might influence cellular response to the treatment. Our findings show that RB becomes dispensable in regulating the cell cycle of NB cells when MYCN is amplified. The described genetic interaction among MYCN, RB1, and E2F3 provides the rationale for the employment of cyclin/CDK complex inhibitors in those NBs carrying *MYCN* amplification and relatively high levels of RB1 expression.

## 2. Results

### 2.1. High Expression of E2F3 Correlates with Poor Prognosis despite RB1 Levels in NB Patients

To shed light on the functional interaction between E2F and MYCN, we first analyzed the mRNA expression correlation between all the E2F factors 1 to 8 and MYCN (Appendix A) in a cohort of 152 TARGET-NBL patients [26]. The analyses show that E2F activators’ (E2F1,2,3) expression strongly correlates with MYCN expression with high statistical significance (Figure 1A and Appendix A). However, E2F3 showed the highest correlation with poor prognosis independently from the MYCN levels (Figure 1A and Appendix A). Interestingly, high expression of E2F3 correlates with poor prognosis, independently (Appendix A). Given the importance of E2F3 in NB prognosis, we asked whether high expression of RB1, the E2Fs’ physiological inhibitor, correlates with a better prognosis in patients despite high levels of E2F3. Surprisingly, we did not observe any relevant improvement in the survival rate of patients in the condition of high-E2F3/high-RB1 Figure 1B, top-right box) compared to those in the condition of high-E2F3/low-RB1 (Figure 1B, top-left box), as if E2F3 high levels are epistatic to the function of MYCN and RB1. This finding suggests that RB function might be overcome in NB tumors with high levels of MYCN and E2F activators.

### 2.2. MYCN Overexpression Overcomes RB Repressive Activity

Several studies in rodent models showed cooperation between MYCN and loss of RB1 in accelerating retinoblastoma formation [27]. Strikingly, that effect can also be observed in tumors carrying *MYCN* amplification in which RB1 is not mutated. For instance, a small but significant percentage of retinoblastoma cases are genetically marked by *MYCN* amplification in a wild-type RB1 background [28,29]. This last evidence, together with the correlation between E2F3 expression and the poor outcome in NB patients despite RB1 levels, lets us hypothesize that a high dosage of MYCN in the cell may overcome RB function to promote an E2F-driven transcriptional activation.

To test this hypothesis, we evaluated the activity of the regulatory region of CDC6 human gene, a well-known E2F-responsive promoter [30], as a function of the modulation of the RB and MYCN expression (Figure 2A). Specifically, we engineered the MYCN non-amplified NB cell line SK-N-AS to conditionally knock down (cKD) RB1 through a doxycycline-inducible CRISPRi system (SK-N-AS cKD RB1) (Appendix A). MYCN levels were, instead, modulated by simultaneously transfecting a MYCN-expressing plasmid inside SK-N-AS. We then assessed the activity of the CDC6 promoter in the presence or absence of MYCN over-expression by a dual-luciferase gene reporter assay. Results show that the CDC6 promoter was significantly upregulated (49.4% increase) when RB1 was knocked down (plus DOXY) in the absence of MYCN over-expression. Interestingly, when MYCN was transfected in a condition of endogenous RB1 expression (minus DOXY), we roughly observed the same increase in luciferase activity as when RB1 was KD in a minus MYCN condition (EV). On the other hand, unlike the control condition with low MYCN level (EV), no significant effects on the CDC6 promoter were observed in cells where MYCN was overexpressed and RB1 was knocked down. Taken together, results suggest that RB1 expression status appeared inconsequential on CDC6 promoter activity when MYCN is overexpressed.

In a complementary experiment, we assessed the repressive activity of RB in the presence or absence of high MYCN level. Thus, we used TET21/N, a human NB cell line derived from the MYCN-nonamplified cell line SHEP, carrying the MYCN transgene under the control of a TET-OFF control system and commonly used to study MYCN-dependent mechanisms [31,32,33,34]. TET21/N cells were transfected with pGL3b_EV control, or the CDC6 luciferase construct (pGL3_CDC6), along with E2F3A (the longest isoform encoded by the E2F3 gene) alone and E2F3A + RB wild-type (RB^wt^). Moreover, we also evaluated E2F3A activity, when cells were co-transfected with an RB’s hyperactive mutant (RB^ΔCDK^), in which the 15 Ser/Thr residues were all substituted with alanines to impede RB1 phosphorylation by the cyclin/CDK kinases [35] (Figure 2C). Analyzing the high (-doxy) or the low MYCN (+doxy) condition separately, the co-transfection of the CDC6 promoter together with E2F3A induced a strong upregulation of the luciferase activity compared to the condition of the CDC6 promoter alone. Introducing RB^wt^, we observed the repression of E2F3A activity, and this effect was even stronger with RB^ΔCDK^. However, when introducing the E2F3 + RB^wt^ samples in the ± doxy experimental conditions, we observed a significant decrease in E2F3-driven activity in the absence of MYCN, compared to the +MYCN condition (Figure 2C). Taken together, these findings suggested that MYCN somehow induces RB inactivation, thus overcoming RB activity in blocking the E2F3 function.

### 2.3. MYCN Overcomes RB Function by Upregulating CCNE1/2 and Downregulating CDKN1A

Understanding how MYCN overcomes RB function might lead to new molecular targets for high-risk NB. Groundbreaking studies on MYC, the MYCN paralogue, shed light on its relevant role in cell cycle progression by upregulating cyclin genes and downregulating CDK inhibitors [36,37,38]. However, little is known about the role of MYCN in this context.

Based on that, we hypothesized that MYCN might abolish the RB1 function by modulating its phosphorylation status (RB-PS). Thus, we analyzed the correlation of MYCN expression with that of several genes involved in the cyclin/CDK pathway during the G1/S transition in the NB patients’ RNAseq database [26]. Our analyses show that cyclins E (CCNE1/2) (Figure 3A) positively correlate with MYCN expression (R= 0.37 and 0.26, respectively), whereas the CDKN1A gene encoding for cyclin/CDK inhibitor p21 negatively correlates with MYCN (R= −0.41) (Figure 3A). This latter finding was consistent with previous results showing that MYCN directly represses CDKN1A expression in NB cells [39].

This finding suggests that RB-PS might be affected by *MYCN* amplification generating an aberrant transcriptional upregulation of CCNE1/2 and downregulation of CDKN1A. For this purpose, we analyzed the RB-PS in TET21/N cells in high and low MYCN status (−/+ doxy). Considering that in physiological conditions RB is phosphorylated during the G1/S transition and remains in this state for the rest of the cell cycle, we also assayed the RB-PS in cells synchronized in the G1 phase by serum withdrawal. RB phosphorylation status was determined by immunoblot, using a specific antibody recognizing Thr 821 and Thr 826 phospho-residues, associated with RB1 inactivated status [8,40]. TET21/N cells were synchronized after 18 h of serum starvation both in high and low MYCN conditions. The correct synchronization of TET21/N cell line was evaluated by propidium iodide staining and flow cytometry, observing a higher frequency of cells carrying 2n amount of DNA (cells synchronized in G0/G1 phase of the cell cycle) following serum starvation (Figure 2B, left, Appendix A). As shown in Figure 3B (right), RB is phosphorylated only in high MYCN when cells are synchronized in the G0/G1 phase of the cell cycle (−FBS). Interestingly, no substantial change in RB-PS was observed in asynchronous cells (+FBS) in the presence or absence of MYCN. Consistently with Pearson’s correlation analysis (Figure 3A), G0/G1 synchronized cells showed a significant upregulation of CCNE1/2 and downregulation of CDKN1A (Figure 3C, Appendix A).

### 2.4. Endogenous RB Replacement with RB^ΔCDK^ Triggers Proliferation Arrest in MYCN-Amplified NB Cell Lines

To further corroborate the hypothesis that MYCN can induce RB phosphorylation and its subsequent inactivation, we assessed the cell proliferation phenotype of *MYCN*-amplified cells in which endogenous RB was replaced with its hyperactive mutant (RB^ΔCDK^). For this purpose, we engineered CHP-134 and kelly NB *MYCN* amplified cell lines to cKD RB1 by CRISPRi. First, we stably integrate into the genome the constructs containing the doxycycline-inducible dCas9-KRAB-MeCP2 and the constitutive sgRNA targeting RB promoter. Therefore, we transduced these cell lines with a third construct in which the tetracycline response element (TRE) was inserted upstream of the RB^wt^ or RB^ΔCDK^ cDNAs. A third cell line was used as a control which carried just the empty vector with the TRE element (Figure 4A). With this approach, we were able to repress the endogenous RB1 and modulate the expression of the exogenous RB1^wt^ or RB1^ΔCDK^ by simply adding doxycycline to the culture media. We measured cell growth for 72 h. Results show that knocking down RB as well as overexpressing it in its wild-type form does not relevantly affect cell proliferation (Figure 4B,C). These findings are consistent with the observation that the wild-type RB1 function is not relevant for cell proliferation when MYCN is amplified. However, when the endogenous RB expression was replaced by that of the ΔCDK mutant, we observed a drastic decrease in cell growth (Figure 4B,C). 

### 2.5. RB Expression Influences MYCN-Amplified Cell Line Susceptibility to CDK4/6 Inhibitors

The use of CDK4/6 inhibition has been effectively employed in the therapy of specific adult breast cancer subtypes [41]. Regarding childhood cancers, the use of CDK4/6 inhibitors has been applied only once on a small cohort of young patients affected by different types of tumors such as neuroblastoma, MRT, and rhabdomyosarcoma. Despite the fact that patients enrolled in the therapeutic protocol were not stratified on cytogenetic criteria, trial results appeared satisfactory, particularly for the sub-cohort of neuroblastoma patients for which seven out of the 15 tested patients displayed stable disease [42]. 

Based on that, we speculated that in the context of amplified MYCN, a positive RB1 expression might positively sensitize neuroblastoma cells to CDK4/6 inhibitors. To prove this hypothesis, we employed the CDK4/6 inhibitors palbociclib and ribociclib on CHP-134 and kelly cell lines cKD for RB1. Thus, we treated these cell lines with ribociclib and palbociclib ranging from 15 nM to 10 µM concentrations, and assessed cell viability through MTS assay after 72 h treatment in the presence or absence of doxycycline. Notably, CHP-134 increased resistance to both ribociclib and palbociclib when RB1 was downregulated (Figure 5A). The same resistant phenotype was observed in KD RB1 kelly cells in the presence of doxycycline (Figure 5B). As experimental control, we performed the same experiment on these cell lines expressing a single-guide scrambled RNA (sg scr) to evaluate the potential effects of doxycycline on drug response. As expected, no change in drug susceptibility was observed (Appendix A).

## 3. Discussion

Despite the recent advances in understanding NB biology and new insights into NB vulnerabilities, there is still a substantial unmet medical need for the treatment of high-risk cases. According to international guidelines [43], high-risk patients are treated with a rigorous chemotherapy regimen of cisplatin, vincristine, carboplatin, etoposide, and cyclophosphamide (COJEC). When possible, resection surgery is performed with myeloablative therapy, hematopoietic stem cell reinfusion, and local radiation therapy [44]. However, when these medications are unsuccessful, the rapid development of precision and personalized medicine raises the opportunity to treat high-risk patients according to the molecular features of their tumors. The primary limit in following this approach is NB’s extreme genetic and epigenetic heterogeneity, leading to an intricate matrix of possible approaches [1]. 

This work aims to shed light on how MYCN influences the proper working of the cell cycle machinery, particularly in the G1/S transition, to find molecular targets specific for *MYCN*-amplified NBs. *MYCN* amplification is a widely recognized negative prognostic factor for NB high-risk cases, leading to advanced tumor stage, high aggressiveness, and poor outcome [2]. Recent findings regarding other types of pediatric tumors such as retinoblastoma have shown that inactivation of the RB1 checkpoint is not per se necessary for the progression of cancer. Indeed, *MYCN* amplification has been found as a significant prognostic marker in a small sub-cohort of retinoblastomas carrying functional wt copies of the RB1 gene. This finding suggests that the simple *MYCN* amplification may be sufficient to bypass the checkpoint functions of a wt RB1. The mechanism through which this occurs remains elusive. 

Results herein presented show that altered expression of MYCN can affect the expression of E2F factors with a special impact on that of E2F3. High MYCN levels do not only contribute to increasing E2F3 expression but most importantly modulate the overall phosphorylation of RB. Indeed, when cells are synchronized at the G1/S border by serum starvation, high MYCN levels keep prolonging or stabilizing RB phosphorylation, thus limiting or even preventing RB control on E2F3. This condition results in an increased mRNA expression of those genes that are targeted by E2F3 and involved in G1/S phase progression, such as cyclin D2, other E2Fs, CDK4/6, and CDC2 [31,45,46]. Cdk4/6- cyclin D, and then, later in G1 Cdk2-CycE, are the main cell cycles responsible for RB phosphorylation and its subsequent inactivation. Consistently with that, we also found that *MYCN* upregulates the activity of cyclin/CDK complexes by positively regulating the expression of cyclins E1/2, and negatively regulating the CDK inhibitor CDKN1A (p21). Based on that, our mechanistical proposal is that the high expression of cyclins E1/2 and the low expression of p21 induced by high MYCN expression in the context of *MYCN* amplification might be sufficient for CDK4/6 activation, inducing the subsequent unconditional phosphorylation and inactivation of RB. This putative mechanism leaves E2F free to upregulate genes involved in the S-phase transition, thus eliminating the G1/S checkpoint of the cell cycle.This finding would logically imply that a complete loss of RB function may be instrumental to the potentiation of MYCN-driven oncogenesis. Surprisingly, when we tested MYCN function in cells in which endogenous RB1 expression was replaced with that of a non-phosphorylatable RB^ΔCDK^ protein, cell growth drastically dropped down. A possible explanation for that is that, under those conditions, the non-phosphorylatable RB^ΔCDK^ protein works as a dominant negative factor that sequesters the entire pool of E2F transcription factors, thus blocking cell-cycle progression. Interestingly, when MYCN is high, the simple downregulation of RB1 is ineffective on cell growth (Figure 4 panels B,C) or on reduction of cancer aggressiveness as implied by the Kaplan Meier graphs in which it is shown that E2F3 and MYCN expressions are strong prognostic markers of clinical outcome independently of RB1.

If on one side the expression of wt RB1 can be easily overcome by amplified *MYCN*, on the other side some level of wt RB1 expression is a crucial requirement for the efficacy of CDK4/6 inhibitors. Indeed, in *MYCN*-amplified cells, the presence of a wt RB make them sensitive to palbociclib and ribociclib, two recently developed CDK4/6 inhibitors. When the same cells are, in fact, KD for RB1 expression, they immediately become more resistant to such inhibitors. In this context, the role of CDK4/6 inhibitors palbociclib and ribociclib might act on the G1/S checkpoint activating the braking role of RB, which seems necessary to slow down the cell-cycle machinery and mediate the efficacy of these two drugs.

The findings presented in this work are representative of in vitro studies, focused on just one of the potential mechanisms driving CDK4/6i resistance in NB. In fact, several features driving tumor resistance to this class of drug have been previously described for other cancers, such as CDK genes amplifications and hyperactivation of the PIK3/AKT/mTOR pathway (reviewed in [47]). For this reason, several clinical trials explored the efficacy of different combinations of CDKi with PI3K and mTOR inhibitors to overcome other possible mechanisms that might interfere with the efficacy of CDKi. Examples of drugs employed in these clinical trials were alpelisib and everolimus for the treatment of subtypes of breast cancer, obtaining satisfactory results [48,49]. Thus, the efficacy of CDK4/6 inhibitors in NB might depend on other factors that should be further investigated in an in vivo experimental setting. Moreover, given the central role of E2F in NB biology highlighted in this and previous works [12,13], the efficacy of drugs that directly target E2F activity could be explored. Among the molecules showing a promising effect on E2F inhibition, HLM006474 showed interesting features that could also be investigated in NB [50]. The biological role of HLM006474 has been recently elucidated, mimicking RB overexpression in cell proliferation phenotype in human embryonic stem cells [51], and inducing apoptosis in vitro in melanoma cell lines [52]. Another emerging molecule that might directly affect E2F3 expression is the tumor-suppressor micro RNA miR-448, which has been shown to downregulate E2F3 in colorectal cancer cell lines, inhibiting cell proliferation and inducing apoptosis in this experimental setting [53]. In this last work, miR-448 mimics have been employed, and future experiments in NB could contribute to increasing the knowledge of E2F3 biology in this context, providing new shreds of evidence for NB treatment.

Overall, the results in this study showed support for the employment of CDKi in NB as a new possible strategy of treatment. This possibility has just begun to be explored by us and other researchers. Pre-clinical studies have already demonstrated the promising effects of CDK4/6i in NB. Coherently with our study, this effect was stronger in *MYCN*-amplified cell lines compared with MYCN-nonamplified cell lines. However, no mechanistical explanation was addressed [54]. Here, we propose that the antineoplastic efficacy of CDK inhibitors like palbociclib and ribociclib in the treatment of *MYCN*-amplified neuroblastoma may require some level of expression of the wt RB protein and that testing the expression of that in patients’ tumor samples at the stage of diagnosis may be of some help in designing more precise therapeutic protocols.

## 4. Materials and Methods

### 4.1. Bioinformatic Analyses

Gene expression and clinical data regarding 152 neuroblastoma patient samples (TARGET-NBL program) were downloaded from Xena Browser [26,55]. FPKM-UQ normalized expression data were used. For each paired gene expression plot, four categories were defined as follow: category I represents samples with an expression of “x-axis” gene and “y-axis” gene above the median of all samples; category II represents samples with an expression of “x-axis” gene above the median and “y-axis” gene below the median of all samples; category III represents samples with an expression of “x-axis” gene and “y-axis” gene below the median of all samples; category IV represents samples with an expression of “x-axis” gene below the median and “y-axis” gene above the median of all samples. Survival analysis was performed using “survminer” (v 0. 4.9) R library. Data handling and plotting were performed using “tidyverse” (v. 1.3.2) R library. 

### 4.2. Cell Culture

TET21/N and SK-N-AS were cultured in Dulbecco Modified Eagle Medium (DMEM) (Sigma-Aldrich, Darmstadt, Germany), CHP-134 and kelly were cultured in RPMI 1640 (Sigma-Aldrich). Both media were supplemented with 10% FBS (Gibco, Waltham, MA, USA) and 1% Pen Strep (Sigma-Aldrich), and 1% L-glutamine was also added to RPMI 1640. Cells were grown in a humified incubator at 37 °C with 5% CO_2_. All the cell lines were obtained from ATCC and regularly tested for mycoplasma contamination.

Stable P.B. TRE dCas9-KRAB MeCP2 cell lines were produced by co-transfection of Super Piggy Bac Transposase and Piggy Bac Vector (System Biosciences, Palo Alto, CA, USA) with a 1:5 ratio, using effectene (QIAGEN, Hilden, Germany) as transfection reagent, following manufacturer instructions. Positive cells were selected with hygromycin B (Invitrogen) at 800 µg/mL until complete death was observed in a non-transduced control plate. 

Lentiviruses for sgRNA were produced in HEK-293 packaging cells by co-transfection of pLenti-sgRNA (Addgene plasmid #71409), helper plasmids psPAX2 (Addgene plasmid #12260), and pMD2.G (Addgene plasmid #12259). After 72 h, viral media were collected and used to transduce P.B. TRE dCas9-KRAB MeCP2 cell lines with a MOI of 0.5. The transduction agent used was polybrene (Santa Cruz Biotechnology, Dallas, TX, USA) as 10,000×. Positive cells were selected with puromycin (Sigma-Aldrich) at 1 µg/mL for 7 days. The same protocol was used to produce stable cell lines for pLVX-TRE3G (Takara Bio, Kusatsu, Japan) constructs (EV, RB^wt^, RB^∆CDK^). After transduction, positive cells were selected with geneticin (Gibco) 800 µg/mL until complete death was observed in a non-transduced control plate. 

### 4.3. Cloning and Plasmids

pGL3_CDC6 construct was obtained by directional cloning of the CDC6 gene promoter (892 bp) in the pGL3basic plasmid (Promega, Cat no. E1751). The sequence was amplified through PCR from genomic DNA using the following primers: FW cloning 5′-atatGGTACCGGTTGAGCATTAGAGAGGTAAGG-3′ and RV cloning 5′-gcatAAGCTTGTTCTTTCCGCCCCTGC-3′. The plasmid was digested with Kpn I and Hind III restriction enzymes.

For the generation of the constructs encoding the sgRNA, two complementary oligonucleotides were annealed, phosphorylated, and cloned into the pLenti-sgRNA plasmid digested with BsmBI. The oligonucleotide sequences were: 

RB1sg1_Fw: 5′-caccgCTGAGCGCCGCGTCCAACCG-3′, RB1sg1_Rv: 5′-aaacCGGTTGGACGCGGCGCTCAGc-3′;sg_scrambled_Fw:5′-caccgGCTTAGTTACGCGTGGACGA-3′, sg_scrambled_Rv: 5′-aaacTCGTCCACGCGTAACTAAGCc-3′. For the generation of the TET-ON stable cell line expressing RBwt or RB^ΔCDK^, the sequences to clone were amplified from pCMV HA hRB-wt and pCMV HA hRb delta CDK using the following primers: BamHI_Rb_Fw: 5′-CGAGGATCCATGTACCCATACGATGTTCCA-3′; and NotI_Rb_Rv: 5′-CCCGCGGCCGCGGTACCTCATTTC-3′. Then, the PCR product was digested using BamHI and NotI and cloned into the pLVX-TRE3G (Takara Bio, Cat n. 631193) plasmid in which the puromycin resistance cassette was replaced with a Geneticin-resistant cassette.

pCMV HA hRb delta CDK was a gift from Steven Dowdy (Addgene plasmid # 58906; http://n2t.net/addgene:58906, accessed on 9 march 2023; RRID:Addgene_58906). pCMV HA hRB-wt was a gift from Steven Dowdy (Addgene plasmid # 58905; http://n2t.net/addgene:58905 accessed on 9 march 2023; RRID:Addgene_58905). PB-TRE-dCas9-KRAB-MeCP2 was a gift from Andrea Califano (Addgene plasmid # 122267; http://n2t.net/addgene:122267; RRID:Addgene_122267). pLenti-sgRNA was a gift from Eric Lander & David Sabatini (Addgene plasmid # 71409; http://n2t.net/addgene:71409 accessed on 9 march 2023; RRID:Addgene_71409).

### 4.4. Dual Luciferase Assay

Luciferase reporter activity was measured using the Dual Luciferase Assay System (E1980, Promega, Madison, WI, USA). Chemiluminescence values for Firefly (Photinus pyralis, Pp) and Renilla (Renilla reniformis, Rr) luciferases were measured using a GloMax 20/20 instrument (Promega). TET21/N and SK-N-AS RB1 cKD were treated for 48 h with 1 µg/mL doxycycline. TET21/N (50.000) and SK-N-AS (100.000) were seeded in a 24-well plate, both “–“ and “+” doxycycline. The following day the cells were transfected in duplicate through lipofectamine 3000 (Invitrogen) following manufacturer instructions. 24 h after transfection, the cells were washed with PBS, Passive Lysis Buffer 1X (PLB) (E194A) was added to each well, in constant agitation for 20 min. 15 µL of cells lysate was collected in a 1.5 mL tube and 35 µL of luciferase assay reagent II (LAR II) were added, then Firefly luciferase activity was read. Afterward, 35 µL of Stop & Glo were dispensed in the tube, and the Renilla luciferase activity was read. Data were represented as ratio of Pp/Rr activities and normalized on pGL3b ratio, relative to the correspondent “–“ or “+” doxycycline condition.

### 4.5. Flow Cytometry

For cell cycle synchronization, after 72 h of treatment with 1 µg/mL doxycycline, or not, 4 million TET21/N cells were seeded in a 10 cm dish in DMEM serum-free media, for 18 h. 

Thus, one million cells were resuspended in 1mL of PBS. Next, 2.5 mL of ethanol 100% was added under continuous agitation to allow the fixation without forming cell clumps. Cells were incubated overnight at −20 °C. Afterward, the cells were centrifuged for 5′ at 300 g, washed with PBS, and then resuspended in 500 µL of PBS supplemented with RNAse A 1 µg/mL (Sigma-Aldrich). Cells were incubated for 30′ at 37 °C and then kept on ice. DNA staining was performed by adding 25 µL of propidium iodide 1 mg/mL (Sigma-Aldrich). Fluorescence-activated cell sorting (FACS) analysis was performed on a CytoFLEX (Beckman Coulter), and the data were analyzed with FlowJo V10 software. Cells singlet were gated as PE area vs. PE width and histograms were generated as frequency we PE A.

### 4.6. Western Blot

Total protein extract was obtained by adding to the cells RIPA buffer supplemented with protease inhibitor (PMSF, Sigma-Aldrich, and cOmplete, Roche, Basel, Switzerland) and phosphatase inhibitor (PhosSTOP, Roche Basel, Switzerland). Protein concentrations were measured by BCA protein assay kit (Pierce Biochemicals Waltham, MA, USA) following manufacturer instructions and 50 µg of protein extract was electrophoresed on an 8% polyacrylamide gel and transferred on a nitrocellulose membrane (GE Healthcare Chicago, IL, USA). Primary antibodies: RB (Abcam, ab181616), Phospho-RB (SantaCruz, sc-271930), E2F3A (SantaCruz, sc-56665), MYCN (SantaCruz, sc-53993), HA (Cell Signaling, #3724), GAPDH (10494-1-AP ProteinTech, Rosemont, IL, USA), β-tubulin (ProteinTech 66240-1-IG), β-actin (Sigma-Aldrich, A2228). Secondary antibodies: rabbit anti-mouse and goat anti-rabbit HRP (Jackson, 115-035-003, 111-035-144). Membranes were incubated with Clarity Western ECL (BIO-RAD, Hercules, CA, USA) and then scanned with ChemiDoc (BIO-RAD).

### 4.7. RT-qPCR

Total RNA was extracted with TRI Reagent (Sigma-Aldrich), 500 µL were added to a 10 mm dish. Chloroform was added (0.2 mL/mL TRI Reagent), and after phase separation, the supernatant was mixed 1:1 with isopropanol. RNA was precipitated by salting out, and the pellet was resuspended in 50 µL of molecular biology grade water (Sigma-Aldrich). 5 µg of RNA was treated with DNA-free removal kit (Invitrogen). 1 µg of RNA DNA-free undergo retro-transcription into cDNA with iSCript reverse transcription supermix kit (BIO-RAD). Quantitative teal Time PCR was performed using SSOAdvanced Universal SYBR green supermix (BIO-RAD) on 5 µg of cDNA. 

Primer sequences: RB1: Fw: 5′-ACTCCGTTTTCATGCAGAGACTAA-3′, Rv: 5′-GAGGAATGTGAGGTATTGGTGACA-3′, E2F3A: Fw: 5′-ACTGCTAGCCAGCCCCG-3′, Rv: 5′-GGACTATCTGGACTTCGTAGTGCAGC-3′; CCNE1: Fw: 5′-CAGACCCACAGAGACAGCTTG-3′; Rv: 5′-GCTCTGCTTCTTACCGCTCT-3′, CCNE2: Fw: 5′-CGAGCGGTAGCTGGTCTGG-3′; Rv: 5′-GGGCTGCTGCTTAGCTTGTA-3′, CDKN1A: Fw: 5′-CAGACCAGCATGACAGATTTCTAC-3′; Rv: 5′-TGTAGAGCGGGCCTTTGAGG-3′. TBP: Fw: 5′-CCGCCGGCTGTTTAACTTC-3′, Rv: 5′-AGAAACAGTGATGCTGGGTCA-3′.

### 4.8. Growth Curve 

CHP-134 (3000 cells) and kelly (10,000 cells) stable cell lines for pLVX TRE3G EV, RBwt, and RB∆CDK were seeded in triplicate in a 96-well plate. Prior to the analysis, doxycycline 1 µg/mL was added or not to the wells. The cells were incubated at 37 °C and 5% CO_2_ in the Incucyte^®^ S3 (Sartorius, Gottinga, Germany). Each well was analyzed every two hours for 72 h, taking two images at a 10× magnification on the phase contrast channel. Cell numbers were counted by the Incucyte^®^ 2020C analysis software, which determined the object count per mm^2^ per image, normalized on t0.

### 4.9. CDK4/6 Inhibitors

After 72 h of pre-treatment with doxycycline 1 µg/mL, CHP-134 (5000) and kelly (10,000) (both RB1 cKD) were seeded in triplicate in a 96-well plate. Ribociclib (LEE001) and palbociclib (PD-0332991) (MedChemExpress) were resuspended in DMSO and added to the cells at different concentrations (10.935 µM, 3.645 µM, 1.215 µM, 405 nM, 135 nM, 45 nM, 15 nM, 0 nM), equivalent amounts of DMSO were added to each point to avoid DMSO-related cytotoxicity artifacts. After 72 h of drug treatment, cell viability was evaluated through MTS assay (Promega) following the manufacturer’s instructions. Absorbances were detected by a Victor microplate reader at 490 nm. Three independent experiments were performed for each cell line.

## 5. Conclusions

In conclusion, our findings show that RB becomes dispensable in regulating the cell cycle of NB cells when *MYCN* is amplified. However, the described genetic interactions among amplified *MYCN*, wt RB1, and E2F3 interestingly unveil an unexpected molecular vulnerability that poses the basis to effectively employ cyclin/CDK complex inhibitors to treat those NBs carrying *MYCN* amplification and relatively high expression levels of a functional RB.

## Figures and Tables

**Figure 1 ijms-24-05408-f001:**
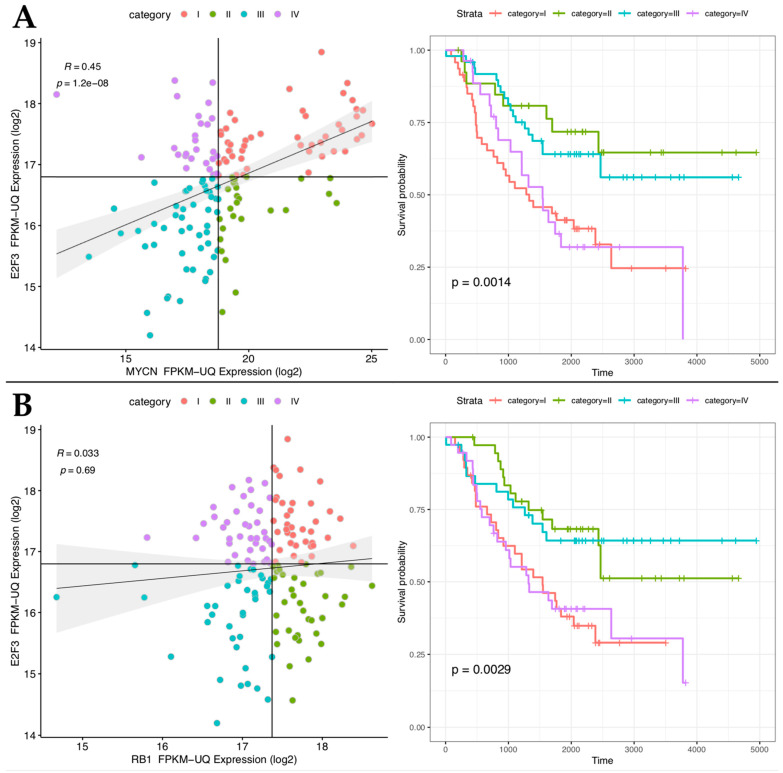
High expression of E2F3 correlates with poor prognosis despite the MYCN and RB1 levels. Left-(**A**): Spearman’s correlation between E2F3 (y-axis) and MYCN (x-axis) in a cohort of 152 target NBL patients. Left (**B**): Spearman’s correlation between E2F3 (y-axis) and RB1 (x-axis) in a cohort of 152 target NBL patients. Correlation coefficient (R) and *p*-value of correlation test are indicated. Right (**A,B**): Overall survival analysis of the target NB dataset for each previously defined category. Correlation coefficient (R) and *p*-value of correlation test are indicated. Samples are divided into four categories as described in Methods (colored as in legend). Colors of Kaplan Meier curves as in legend. The *p*-value of the log-rank test is indicated. *p*-value of proportional hazard assumption was 0.27 and 0.13 for (**A**) and (**B**), respectively.

**Figure 2 ijms-24-05408-f002:**
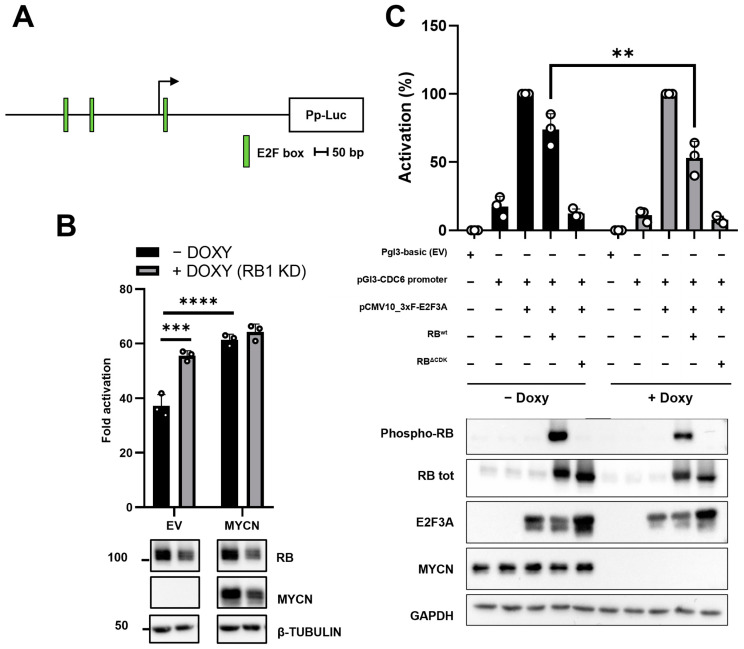
MYCN overcomes RB function in SK-N-AS and TET21/N cell lines. (**A**) Schematic illustration of the CDC6 promoter region. E2F consensus boxes are highlighted in green. Pp: Photinus pyralis luciferase; bp: base pairs (**B**) Up: dual luciferase assay in SK-N-AS cKD RB1 cell line. Cells were incubated with or without doxycycline for 48 h to induce RB1 downregulation, then were transfected with pGL3b, pGL3b_CDC6 + EV, or pGL3b_CDC6 + MYCN. Constitutive promoter-driven Renilla reniformis (Rr) luciferase was used as internal control for transfection efficiency. Fold activation values represent Pp/Rr luciferase activity relative to the pGL3b alone condition. Down: western blot of the same cells transfected with the indicated construct. β-tubulin was used as loading control (**C**) Up: dual luciferase assay in the TET21/N cell line. Cells were incubated with or without doxycycline for 48 h to induce MYCN downregulation, then were transfected as shown. Constitutive promoter-driven Rr luciferase was used as internal control. RB tot = total RB. Activation (%) values represent Pp/Rr luciferase activity relative to pGL3b_CDC6 + E2F3A sample. Down: western blot of the same cells transfected as indicated. GAPDH was used as loading control. Luciferase experiments results were shown as an average of three (N = 3) independent experiments ±SD. **, ***, **** indicated *p* < 0.01, 0.001, and 0.0001, respectively. Statistical analyses were performed with two-way ANOVA.

**Figure 3 ijms-24-05408-f003:**
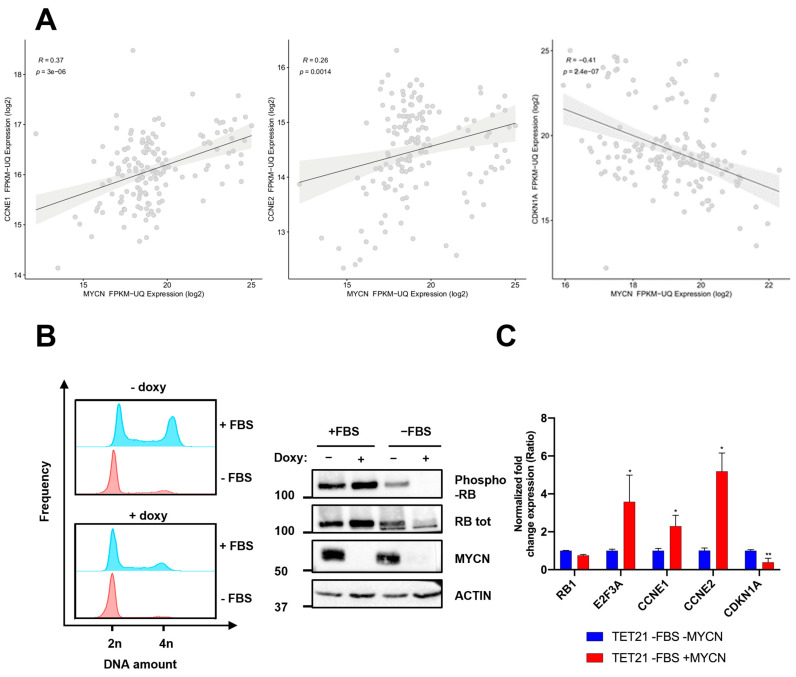
MYCN induces RB phosphorylation in the G1 phase. (**A**) Pearson’s correlation between MYCN and CCNE1/2 or CDKN1A in a cohort of 152 NB patients (NBL dataset). The p-value by the Mann–Whitney test is indicated. (**B**) TET21/N cells were incubated with 1µg/mL of doxycycline for 48 h to induce MYCN KD and seeded at low density in the presence or absence of fetal bovine serum (FBS) for G0/G1 synchronization. Left: After 18 h, cells were harvested and stained with propidium iodide for cell cycle analysis in flow cytometry. 2n and 4n indicate the relative amount of DNA. Right: Western blot of TET21/N cells treated as described. Results are representative of two (N = 2) independent experiments (**C**) qRT-PCR on G1 synchronized TET21/N cells in the presence (-MYCN) or absence (+MYCN) of doxycycline. Data were normalized using the TBP housekeeping gene. The experiment was performed in triplicate (N = 3). Data were plotted as the mean ± SD. Statistical analyses were performed using two-way ANOVA. Error bars represent SD. *, **, indicated *p* < 0.05, 0.001, respectively.

**Figure 4 ijms-24-05408-f004:**
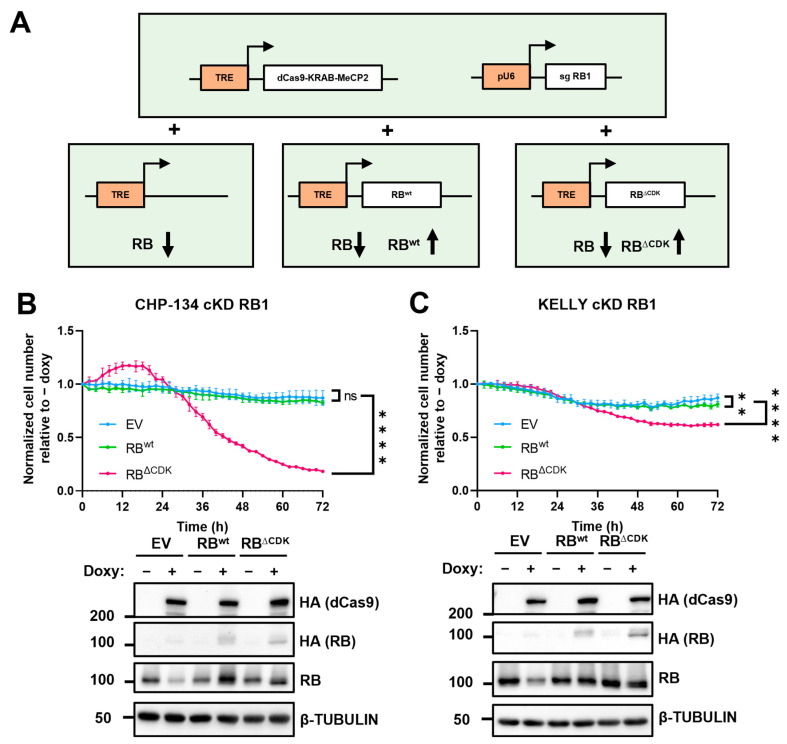
Endogenous RB replacement with RB^ΔCDK^ induces a drastic effect on cell proliferation in *MYCN*-amplified cell lines. (**A**) Schematic illustration of the system. dCas9-KRAB-MeCP2 and RBwt/ΔCDK are under the control of a tetracycline response element (TRE) responsive to the trans-activator rTtA. Constitutive pU6 promoter drives the expression of the sgRNA targeting the RB1 promoter (sgRNA RB1). In the presence of doxycycline, RB1 is repressed and exogenous RBwt/ΔCDK is upregulated depending on the transduced construct. (**B**,**C**) Up: CHP-134 and kelly cKD RB1 EV, WT, or ΔCDK cell lines were seeded in the presence or absence of doxycycline. Microphotography pictures were taken every two hours, and cells were counted by Incucyte2020C software. Results were plotted as a ratio between the number of cells normalized on 0 h of the +doxy sample divided by the number of cells normalized on 0 h of the -doxy sample (Normalized cell number relative to -doxy). Two-way ANOVA multiple comparisons was performed. **, **** indicates *p* < 0.01, 0.0001, respectively at the last time-point; ns indicates *p* ≥ 0.5. Down: Western blot ensuring the proper working of the system.

**Figure 5 ijms-24-05408-f005:**
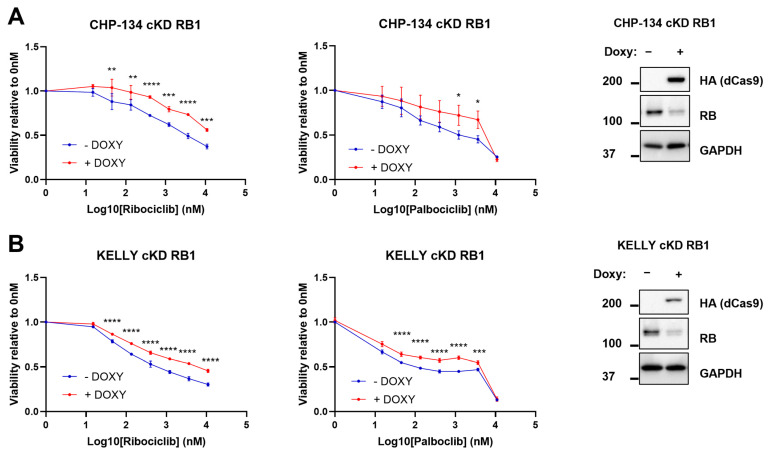
RB1 downregulation induces resistance to CDK4/6 inhibitors ribociclib and palbociclib. CHP-124 and kelly cKD RB1 cells were seeded at low confluence and treated with ribociclib or palbociclib for 72 h in the presence or absence of doxycycline. (**A**) Left: MTS results of CHP-134 cKD RB1. Right: western blot confirming the correct downregulation of RB1. (**B**) Left: MTS results of kelly cKD RB1. Right: western blot confirming the correct downregulation of RB1. Data are plotted as absorbances normalized on the 0nM sample as averages of three (N = 3) independent experiments. Error bars represented SD. Statistical analysis was performed by two-way ANOVA comparisons between − doxy and + doxy samples per each concentration. *, **, ***, **** indicated *p* < 0.05, 0.01, 0.001, and 0.0001, respectively.

## Data Availability

Publicly available gene expression and clinical data regarding neuroblastoma patient samples (from TARGET-NBL program) were analyzed in this study. This data were downloaded from Xena Browser at the following link: https://xenabrowser.net/datapages/?cohort=GDC%20TARGET-NBL&removeHub=https%3A%2F%2Fxena.treehouse.gi.ucsc.edu%3A443 accessed on 1 September 2022.

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
