# Peer review of "MYCN Amplification, along with Wild-Type RB1 Expression, Enhances CDK4/6 Inhibitors’ Efficacy in Neuroblastoma Cells"

_ijms, 2023, doi:10.3390/ijms24065408_

Round 1

Author Response

Before replying to the reviewer’s comments, we wish to express our thanks to him/her for the constructive criticisms and helpful suggestions that we took into consideration to improve the quality and significance of the manuscript.

You’ll find attached a file word with detailed answers formatted as follow:

Reviewer questions are indicated in lines with “-” and our answers are reported in italics.

Reviewer 2 Report

This article identified that MYCN-amplified cells have RB inactivation by inducing RB hyperphosphorylation during the G1 phase, which is novel and meets the scope of IJMS. However, additional work needs to be done.

MYCN should be in italics if presenting about amplification. 

Authors cannot have a title with “vulnerability window for CDK4/6 inhibitors” if not tested comprehensively in vitro and in vivo in patient-derived MYCN-amp cell lines.

Is the log-rank test p-value adjusted for multiple comparisons? Did the authors test the proportional hazard assumption? To define E2F3's independent role in prognosis, authors should perform univariate and multivariate analysis.

RB phosphorylation should be shown in some patient-derived MYCN-amp and non-MYCN-amp NB cell lines.

What is the mechanism of RB phosphorylation in MYCN-amp NB cells?? At least need to be discussed.

The discussion seems like a reiteration of results. Discussion should be well organized by briefly mentioning findings, limitations, future work, and unanswered questions.

Doxycycline can change drug response; appropriate controls are needed for figure 5.

Where is the comparison of cytotoxicity in MYCN-amp vs. non-MYCN-amp cell lines for CDK4/6 inhibitors?? Authors should also consider using MYCN overexpression cells to show more activity for CDK4/6 inhibitors in MYCN amplified cells. 

Author Response

(The authors gave the same response as above.)

Reviewer 3 Report

Reviewer’s report

MS: ijms-2146754

This study aimed to identify an actionable target of undruggable MYCN via the cell cycle checkpoint machinery, and proposed CDK inhibitors for treating a subset of NB patients with MYCN amplification and a relative RB1 high expression. This study could be important. Please find below my comments which I believe would help improved the quality of the manuscript.

Major comments

1.     The effects of RB1 KD in MYCN-amp NB cells in vitro on CDK4/6 inhibitor resistance, even though statistically significant, seemed to be marginal. Without more supportive evidence from 3D culture/in vivo models, the impact of this mechanistic finding could be limited. This limitation should be carefully discussed.

2.     Identifying E2F3 correlation with poor prognosis despite the MYCN and RB1 status is interesting as the direct inhibition of E2F3 probably have a therapeutic potential for high-risk NB, for example, by HLM006474, a small molecule pan-E2F inhibitor, and miR-448 mimics. In my view, therapeutic strategies that directly target E2F3 should also be discussed, considering the numbers of patients with MYCN-amp and high expressions of RB1 and E2F3, in the TARGET cohort used in this study.

Minor comments

1.     The method to define the categories I-IV in Figure 1 should be described in the main text (e.g., the method section), not just in the figure legend.  

2.     The bar colors in Figure 2B and 2C should be the same system (e.g., grey for + Doxy) to avoid confusion.

3.     The subheading 2.3 “MYCN overcomes RB function by upregulating and downregulating CDKN1A, respectively” was confusing. Please revise.

Author Response

(The authors gave the same response as above.)

Round 2

Reviewer 1 Report

The reviewer is satisfied with the author’s adjustments to the manuscript.

Author Response

Dear Reviewer, we are happy to see that you were satisfied with our replies and we thank you again for your helpful suggestions and critical comments.

Best regards